# Social jetlag and sleep debts are altered in different rosters of night shift work

Swaantje Casjens[1]*, Frank Brenscheidt[2], Anita Tisch[2], Beate Beermann[2], Thomas Brüning[1], Thomas Behrens[1], Sylvia Rabstein[1]

1 Institute for Prevention and Occupational Medicine of the German Social Accident Insurance, Institute of the Ruhr University Bochum (IPA), Bochum, Germany, 2 Federal Institute for Occupational Safety and Health (BAuA), Dortmund, Germany

* Swaantje.Casjens@ruhr-uni-bochum.de

**Data Availability Statement:** The study was funded by the Federal Institute for Occupational Safety and Health (Project No.: F 2409). This does

## Abstract

### Background

Night and shift work are suspected to cause various adverse effects on health and sleep. Sleep deprivation through shift work is assumed to be compensated on free days. So far it is not clear how different shift systems and shift lengths affect sleep structure on work and free days. Especially working night shifts disrupts the circadian rhythm but also extended working-hours (12h) might affect sleep characteristics. Hitherto, the magnitude of sleep debt, social jetlag, and Locomotor Inactivity During Sleep (LIDS) in different shift systems is unknown.

### Methods

Here, we investigated the impact of five different shift rosters on sleep in 129 industrial workers from Germany. Permanent night work with multiple shift systems with and without night shifts and with different shift lengths were compared. Wrist-activity was monitored over 28 days revealing sleep on- and offsets as well as LIDS as proxy for sleep quality. Overall, 3,865 sleep bouts comprising 22,310 hours of sleep were examined.

### Results

The mean daily age-adjusted sleep duration (including naps) was 6:43h and did not differ between shift workers of different rosters. However, sleep duration on workdays was particularly low in rotational shift systems with 12h-shifts (5:00h), while overall sleep debt was highest. Shift workers showed a median absolute social jetlag of 3:03h, which differed considerably between shift types and rosters (p<0.0001). Permanent night workers had the highest social jetlag (5:08h) and latest mid-sleeps on workdays and free days. Sleep quality was reduced in permanent night shift workers compared with shift workers in other rosters and differed between daytime and nighttime sleep.

not alter our compliance with the PLOS ONE policies for sharing data and materials. The data that support the findings of this study cannot be shared publicly. However, data are available at the Federal Institute for Occupational Safety and Health upon reasonable request (Federal Institute for Occupational Safety and Health (BAuA), Friedrich-Henkel-Weg 1-25, 44149 Dortmund, Germany; email Fb1.1@baua.bund.de).

**Funding:** The study was funded by the Federal Institute for Occupational Safety and Health (Project No.: F 2409). The Open Access Publication Fund of the Ruhr-Universität Bochum contributed to publication costs for this manuscript. The funders had no role in study design, data collection and analysis, decision to publish, or preparation of the manuscript.

**Competing interests:** The authors have declared that no competing interests exist.

## Conclusions

Shift work leads to partial sleep deprivation, which particularly affects workers in 12h-shifts and permanent night shifts. Working these shifts resulted in higher sleep debts and larger absolute social jetlag whereas sleep quality was especially reduced in permanent night shift workers compared with shift workers of other rosters.

## Introduction

One of the key health consequences of shift work is sleep loss which has been the interest of systematic research for more than thirty years [1]. Especially night shifts forcefully disrupt the circadian rhythm, leading to shorter sleep, increased sleepiness, an accumulation of sleep debt, and fatigue [2]. The uncertainty about the magnitude of sleep debt and social jet lag in different rosters makes it currently difficult to reorganize shift systems. It has been shown that day sleep after night shifts [3] as well as main sleep before early morning or night shifts is often short-ened [4]. Specific shift characteristics, such as quick returns, night or irregular work, and morning preference were also associated with shorter sleep duration, poorer sleep quality, and disturbed sleep [5–8]. On the other hand, employees working extended shifts (12h-shifts) might sleep longer and better than employees of 8h-shift schedules because the increased number of work-free days may improve psycho-physical recovery, family and social life [9]. However, doubled risks of occupational accidents towards the end of a 12h-shift compared to 8h-shifts have been reported [10].

Adequate sleep is essential to maintain physical and mental health. Previous meta-analyses have shown, that short sleep and poor sleep quality can lead to immunological changes, increase the risk of cognitive impairments, obesity, type 2 diabetes, coronary heart disease, stroke, and depression [1] as well as to an increase of all-cause mortality [11]. Furthermore, meta-analyses found associations between sleep quality and occupational accidents and injuries, and falls [12, 13] as well as more sleep disturbances in shift workers than in day workers [14, 15]. More recent studies showed that long-term exposure to shift work with night shifts was associated with a substantial increase of fatigue and long sleep ($\geq$ 9 hours) during work-free days [16] as well as decreased psychomotor vigilance [17]. Especially, permanent night workers reported poorer health, more absenteeism and less job satisfaction than day workers [18, 19] as well as more insomnia on work-free days in comparison to workers in two and three-shift rotations [20]. However, other studies did not find an association between shift work with night shifts and sleep quality [21, 22]. Nevertheless, leaving night work in comparison to staying in such work was associated with improved sleep [22].

The gold standard to assess sleep quality is laboratory polysomnography (PSG) being an all-embracing but time-consuming and expensive technique. However, application of PSG is not feasible in epidemiological studies with many participants. Alternatively, locomotor activity can be recorded continuously with small devices (actigraphy) at home allowing to calculate sleep-wake scores and obtain time and duration of sleep bouts under normal conditions outside the laboratory [23]. With the exception of a few studies [7, 24], sleep in shift workers has often only been investigated in far fewer than a hundred subjects using actigraphy [5, 25] The duration of sleep measurement using actigraphy also varies between studies, ranging from a few days to several weeks. Regardless of the industry examined, all studies observed impaired sleep among shift workers, such as shorter sleep duration, increased sleepiness, more prema-ture awaking or delayed sleep timing. Furthermore, it was proposed to extract locomotor

inactivity during sleep (LIDS) from wrist movements measured by actigraphy [24]. LIDS has been presented to be closely related with PSG-monitored sleep physiology and hence be able to reveal relevant information about sleep structure and sleep quality.

Here, we investigate the impact of different shift rosters on sleep in industrial workers over a study period of 28 days. We analyze overall sleep duration, timing of sleep, sleep debt and naps on workdays and free days, as well as characteristics of LIDS considering different shifts. We compare permanent night work with multiple shift systems with and without night shifts and with different shift lengths and provide the first LIDS analysis of shift workers with different shift types.

## Materials and methods

### Study population

In total, 143 shift workers were recruited from two manufacturing companies with partially rotating shift schedules in Germany. The study period was 28 days for each participant and took place in spring 2018 (company 1) and fall 2018 (company 2). All shift-workers of work areas within the two companies who shared similar work tasks were eligible. All participants provided written informed consent. The study was approved by the Ethics Committee of the Ruhr University Bochum, Germany (Reg. No. 17–6205).

In company 1, shift workers worked in fast-forward rotating 8h-morning, evening, and night shifts each with two equal shifts in succession (transition times: 06:00, 14:00, and 22:00) or in 12h-shifts (transition times: 06:00 and 18:00) with one week consisting of only day shifts followed by a week of night shifts only. Both rosters also included weekend shifts with 42 weekly working hours each. In company 2, three different rosters existed: 1) rotational shift work with morning shifts (06:00 to 14:30) and evening shifts (14:30 to 22:30) without night work and without weekend shifts; 2) 12h-weekend shifts rotating between day and night shift every other weekend (06:00 to 18:00 or 18:00 to 06:00); 3) permanent night work (22:30 to 06:00) without weekend shifts. Detailed shift schedules of the individual rosters are shown in S1 File.

In this study, only participants with recordings of wrist-activity were included (n = 137). Participants were excluded from analyses if they worked less than five days during the study period or worked ten or more additional shifts according to the roster, did not work at least one night shift if night work was part of the roster (n = 3), or if actimetry existed for less than ten days (n = 3). Due to deficient recordings of wrist-activity two participants were further excluded. The remaining study population (n = 129) comprised 90 shift workers in company 1 and 39 shift workers in company 2. For each participant, sociodemographic characteristics, alarm clock use, or other reasons influencing sleep times, e.g. children and medications were assessed by questionnaire. Body height and weight were measured.

### Actimetry data processing

Wrist-activity was monitored using the tri-axial accelerometer ActiGraph GT3X (Actigraph, Pensacola, FL, USA). Activity was sampled in 1s intervals and stored in 30s intervals. Participants wore the devices continuously for four weeks. Additionally, each participant was asked to record information regarding their sleeping and working times and times not wearing the actigraph into a protocol for every day during the study period (sleep diary). With the help of the protocol sick days, holidays, and irregularities in the shift schedule could be identified. Sleep on- and offsets were determined from actimetry data using the algorithm by Cole and Kripke coming with the device's software [26]. Two authors (SC and SR) have cross-checked

all sleep on- and offsets derived from actigraphy and from sleep diaries for each participant for plausibility. Based on expert rating, differences were adjusted or discarded.

The actigraphs did not automatically adjust for summer and winter time. Hence, sleep bouts ending after the time change were manually adjusted. Times were put forward (winter to summer time) respectively backward (summer to winter time) by one hour. Accordingly, sleep bouts at the time change were reduced respectively extended by one hour to analyze the actual sleep duration.

## Primary sleep parameters

Overall, 3,865 sleep bouts comprising 22,310 hours of sleep were examined. Based on the activity-derived sleep data, duration of main sleep (longest sleep of a day), and naps were calculated. We distinguished between restricted sleep episodes at workdays (sleep before morning shifts, after evening or night shifts), unrestricted sleep episodes following shift (sleep on work-free days after morning, evening or night shifts, between morning and evening/night shifts or between work-free days and evening/night shifts), and subsequent free days (sleep from subsequent work-free days) [27]. Sleep debt was assessed as the absolute difference of sleep duration between workdays and work-free days. Sleep starting and ending between 6am and 8pm was defined as day sleep and sleep between 8pm and 8am as night sleep, respectively. Other sleep episodes were assigned to a third group of unclassified sleep.

Mid-sleep was calculated for main sleep episodes only. Shift-workers were chronotyped by the mid-sleep between two work-free days following an evening shift ($MSF^E_{sc}$) corrected for oversleep on work-free days [25]. If the shift schedule did not comprise evening shifts, we applied a linear mixed model to estimate $MSF^E_{sc}$ from work-free days after night shifts [25, 27]. As proposed before [25], we corrected chronotype of the workers from company 2 (+8.98˚) to the longitude of the population in company 1 (+7.09˚) using the following equation:

$$MSF^E_{sc,adj} = MSF^E_{sc} + (8.98 - 7.09) * 4/60.$$

Chronotype was categorized in early, intermediate and late with cutoffs at the first quartile (04:00) and third quartile (04:27).

Social jetlag describing the difference between mid-sleeps on work and work-free days was used as a proxy for circadian misalignment [28]. To compare individual average social jetlag across shift systems, absolute social jetlag across all shifts was calculated as weighted average of the shift-specific social jetlag [6, 25]. Increased social jetlag values indicate a greater disrupted sleep timing between work and work-free days.

Overall sleep duration was calculated as sum of all sleeps only if the participant had worn actigraphs over the complete study period of 28 days.

## Locomotor Inactivity During Sleep (LIDS)

All activity records were evaluated in bins of 10 minutes (min). For each bin i, locomotor activity was calculated as resultant acceleration ($r_i$) of the three acceleration signals in x- y- and z-direction ($r_i = \sqrt{x_i^2 + y_i^2 + z_i^2}$). As proposed by Winnebeck et al., the resultant acceleration signal was inverted non-linearly to inactivity in values from 0 to 100 to enhance the contrast between movement and non-movement during sleep ($LIDS_i = \frac{100}{r_i+1}$) [24]. Afterwards inactivity was smoothed via a 30-min centered moving average. We applied the LIDS conversion to all main sleep bouts (n = 3,465) with a length of 3 to 12 hours (n = 3,331). Individual sleep bout timelines were normalized to their LIDS period determined via cosine model fitting. In brief,

1-harmonic cosine models with periods from 30 to 180 min in steps of 5 min were fitted to LIDS iteratively. The best-fit model was the one with maximal Munich Rhymthmicity Index (MRI = range of oscillation x bivariate correlation coefficient) from which the optimal LIDS period (optPer) was derived. The cosine fit was statistically significant for 97% of bouts. In accordance with an earlier study, all bouts with statistically non-significant cosine model fits (n = 110) were excluded from further analyses [24]. For averaging LIDS across all sleep bouts, external times of bouts were converted to internal times relative to optPer and normalized to 110-min/cycles ($t_{int} = t_{ext}$/optPer x 110 min). We chose a cycle length of 110 min in contrast to a 90-min cycle length as proposed before [24] because here the median of optPer was 110 min (IQR 90–150). Overall, 3,221 out of 3,465 main sleep bouts fulfilled the analysis criteria of duration and quality for LIDS analysis. The difference between the minimum and maximum LIDS level per LIDS cycle (calculated from the raw LIDS values of each cycle) was determined to assess the range of oscillation (= amplitude) per cycle reflecting the strength of the LIDS rhythm. Usually, oscillations with larger amplitudes are more stable than those with lower amplitudes.

## Statistics

Statistical analyses were undertaken using SAS version 9.4 (SAS Institute Inc., Cary, NC, USA) and R version 4.0.3. GraphPad Prism version 7.04 (GraphPad Software, La Jolla California, USA) was used to prepare graphs. Median and inter-quartile range (IQR) were calculated to describe the distribution of continuous variables. Group differences were tested with the Kruskal-Wallis test. Means and 95% confidence intervals (CI) for age-adjusted sleep bout duration and mid-sleeps were assessed with mixed models allowing for repeated measures for subjects with compound symmetry covariance structure. Differences between shift systems were tested with the F test. We fitted linear mixed-effects regression models to LIDS levels and LIDS oscillation with a nested structure. LIDS cycles were nested within sleep bouts which were nested within subjects. As fixed effects, we included shift system, age (per 10 years), sex, sleep duration (h), number of LIDS cycles, and schedule-based sleep characteristics (night sleep *vs*. day sleep, work day *vs*. work-free day). To depict variable-specific declines across LIDS cycles, we modelled interactions of number of LIDS cycles with all included fixed effects. Random intercepts were specified for each nested effect (subjects, sleep bouts, LIDS cycles) and random slopes within each subject accounted for unsystematic variation in LIDS levels and decline. In contrast to the fixed effects, which we expect will have an effect on the dependent variable, random effects are usually grouping factors which we are trying to control for. We have no particular interest in their impact on the response variable. Hence, we focus on the fixed effects parts of the models being comparable to the results of a simple linear model.

## Results

Demographic information on sex, age, body mass index, chronotype, napping behavior, and overall sleep duration during the whole study are shown in Table 1. Most participants were male (90.7%). The median age was 47 years (range 20–64 years). Workers with permanent night shifts were the oldest whereas workers with 8h-shifts with night work had the longest shift work experience. More than every second employee napped at least once on workdays. Permanent night shift workers and workers with 8h-rotating shifts including night work napped the most. The mean age-adjusted chronotype of the overall sample was 4:12 (95% CI 4:00–4:27). The chronotype of shift workers without night shifts was advanced by 51 min compared to the whole study group and by more than 70 min compared to workers with permanent night shifts or weekend shifts only.

**Table 1. Description of the study population.**

| Characteristics | Total | Company 1 | | Company 2 | | | |
|---|---|---|---|---|---|---|---|
| | | 8h-shifts with night work | 12h-shifts (day/night) | 12h-weekend shifts (day/night) | 8h-shifts without night work | Permanent night shifts | P value |
| | n (%) / median (IQR[a]) | n (%) / median (IQR[a]) | n (%) / median (IQR[a]) | n (%) / median (IQR[a]) | n (%) / median (IQR[a]) | n (%) / median (IQR[a]) | |
| n | 129 | 43 | 47 | 11 | 12 | 16 | |
| Men | 117 (90.7) | 41 (95.4) | 45 (95.7) | 8 (72.7) | 11 (91.7) | 12 (75.0) | 0.021[d] |
| Age [years] | 47 (41–54) | 48 (35–52) | 45 (37–49) | 46 (43–54) | 51 (39.5–56) | 52 (48.5–54.5) | 0.028[e] |
| Body mass index [kg/m²] | 28.4 (25.1–31.8) | 28.4 (25.1–33.4) | 27.7 (25.4–31.5) | 26.8 (24.5–30.8) | 29.2 (25.9–29.6) | 29.5 (24.3–32.3) | 0.927[e] |
| Years of shift work[b] | 21 (14–28) | 28 (15–31) | 26 (13–28) | 19 (17–21) | 12.5 (2.5–22.5) | 18 (16–21.5) | 0.010[e] |
| Subjects doing naps | 88 (68.2) | 36 (83.7) | 28 (59.6) | 6 (54.5) | 6 (50.0) | 12 (75.0) | 0.039[d] |
| Subjects doing naps on workdays | 72 (55.8) | 28 (65.1) | 24 (51.1) | 5 (45.5) | 4 (33.3) | 11 (68.8) | 0.207[d] |
| Chronotype[c] [local time] | 4:12 (4:00–4:27) | 4:09 (4:00–4:22) | 4:12 (4:03–4:22) | 4:30 (4:19–4:44) | 3:30 (2:26–3:36) | 4:28 (4:19–4:40) | <0.001[e] |

[a]Inter-quartile range

[b]Years of shift work could not be determined for all 129 subjects

[c]Chronotype was assessed as mid-sleep between two work-free days as described in the method section and could not be determined for seven subjects due to insufficient subsequent free days after shifts (n = 122)

[d]Fisher's exact test

[e]Kruskal-Wallis test.

Average values of the main sleep characteristics (time of mid-sleeps, social jetlag, sleep duration, sleep debt) on workdays, work-free days, and subsequent free days are presented in Tables 2 and 3. Mid-sleeps before morning or after evening shifts and on work-free days after morning and evening shifts were similar across groups (Table 2). However, permanent night-shift workers slept latest before night shifts on workdays, on work-free days after night shifts, and on subsequent free days in comparison to the other shift groups. Shift workers showed a median absolute social jetlag of 3:03 h:min, with highest values regarding night shifts (5:27 h:mm). Social jetlag on different shifts did not differ between shift groups. However, absolute social jetlag was highest in permanent night-shift workers. Furthermore, absolute social jetlag was higher in 12h-shift workers (day/night) than in 8h-shift workers with night work (3:43 h:min vs. 1:48 h:min, p<0.0001).

The mean daily age-adjusted sleep duration (main sleeps and naps) was 6:26 h:min (95% CI 6:18–6:34) and did not differ between groups (Table 3). During workdays, workers with permanent night shifts and workers with 8h-shifts including night work compensated for shorter main sleeps (mean 5:13 h:min, 95% CI 4:51–5:35; mean 5:23 h:min, 95% CI 5:09–5:36) with extensive naps before or after the shifts (mean: 2:15 h:min and 2:55 h:min) in comparison to other study groups. The shortest sleep duration on workdays was observed for workers with 12h-shifts including night work which they compensate with longer sleep during work-free days resulting in a daily sleep debt of more than three hours. The lowest sleep debt was observed in workers without night work (1:24 h:min, 95% CI 0:46–2:02). In the subgroup of 90 shift workers from company 1, the age-adjusted sleep deficit in subjects working extended 12h-shift was higher than in subjects working normal 8h-shifts with night work (p<0.0001).

Sleep quality was assessed according to LIDS levels and LIDS oscillation which were analyzed with linear mixed models (Table 4). Study groups differed in their overall LIDS levels. Workers with permanent night shifts represented lower LIDS levels ($\beta_{\text{Permanent night shifts}} =$

**Table 2. Age-adjusted mid-sleep of main sleep episodes and social jetlag with 95% confidence intervals (CI) in the whole study population and stratified by roster.**

| | Total | 12h-shifts (day/night) | 8h-shifts with night work | 12h-weekend shifts (day/night) | 8h-shifts without night work | Permanent night shifts | F test |
|---|---|---|---|---|---|---|---|
| | | N = 47 | N = 43 | N = 11 | N = 12 | N = 16 | |
| | Mean (95% CI) | Mean (95% CI) | Mean (95% CI) | Mean (95% CI) | Mean (95% CI) | Mean (95% CI) | |
| Workdays | | | | | | | |
| Morning shifts | 1:34 (1:28–1:40) | 1:39 (1:30–1:48) | 1:30 (1:21–1:39) | 1:45 (1:26–2:04) | 1:22 (1:06–1:39) | - | 0.1580 |
| Evening shifts | 4:11 (3:54–4:27) | - | 4:12 (3:53–4:31) | - | 4:07 (3:31–4:43) | - | 0.8124 |
| Night shifts | 9:57 (9:48–10:06) | 9:51 (9:38–10:05) | 9:39 (9:25–9:54) | 10:07 (9:39–10:36) | | 10:41 (10:20–11:02) | <0.0001 |
| Work-free days | | | | | | | |
| After morning shifts | 3:25 (3:12–3:39) | 3:31 (3:10–3:52) | 3:19 (2:55–3:42) | 3:35 (2:50–4:20) | 3:13 (2:28–3:57) | - | 0.7650 |
| After evening shifts | 4:17 (3:40–4:55) | - | 4:06 (3:20–4:52) | - | 4:40 (3:34–5:46) | - | 0.4010 |
| After night shifts | 4:34 (4:11–4:57) | 4:11 (3:40–4:41) | 4:03 (3:05–5:00) | 5:14 (4:06–6:22) | | 5:47 (4:54–6:41) | 0.0121 |
| Subsequent free days | 4:05 (3:52–4:18) | 3:53 (3:31–4:14) | 4:00 (3:40–4:20) | 3:20 (2:42–3:59) | 3:57 (3:17–4:37) | 5:33 (4:58–6:08) | <0.0001 |
| Naps | 15:57 (15:21–16:32) | 14:54 (13:57–15:52) | 15:37 (14:46–16:28) | 16:51 (15:01–18:42) | 16:47 (14:46–18:47) | 17:36 (16:25–18:48) | 0.0091 |
| Social jetlag [h:min] | | | | | | | |
| Morning shifts | 1:53 (1:31–2:04) | 1:55 (1:36–2:14) | 1:50 (1:31–2:10) | 1:51 (1:13–2:29) | 1:54 (1:17–2:30) | - | 0.9380 |
| Evening shifts | 1:07 (0:49–1:26) | - | 1:06 (0:43–1:28) | - | 1:10 (0:38–1:33) | - | 0.8096 |
| Night shifts | 5:27 (5:05–5:48) | 5:40 (5:12–6:07) | 5:27 (4:32–6:22) | 5:00 (3:55–6:04) | | 5:08 (3:57–6:18) | 0.5829 |
| Absolute | 3:03 (2:44–3:21) | 3:43 (3:25–4:02) | 1:48 (1:24–2:11) | 3:36 (2:54–4:19) | 1:34 (0:59–2:09) | 5:08 (3:57–6:18) | <0.0001 |

-10.07 LIDS units, Table 4), especially at the beginning of a sleep bout (Fig 1A). LIDS levels were lower (i.e. more movement) in men ($\beta$ = -7.65 LIDS units). Sleep at nighttime and work-day sleep showed also lower LIDS levels ($\beta_{\text{Night sleep}}$ = -6.78 LIDS, $\beta_{\text{Work day}}$ = -1.29 LIDS units, Table 4). Furthermore, over the course of a sleep bout decline rates in mean LIDS levels differed between groups. Permanent night-shift workers showed a shallower decline than other shift workers ($\beta_{\text{LIDS cycle number*permanent night shifts}}$ = 3.09). A shallower decline of LIDS levels could also be seen for nighttime sleep, workdays, and sleep duration ($\beta_{\text{LIDS cycle number*night sleep}}$ = 1.43, $\beta_{\text{LIDS cycle number*work day}}$ = 0.94, $\beta_{\text{LIDS cycle number*sleep duration}}$ = 0.64). For every hour of additional sleep, rate of decline per LIDS cycle flattens by 0.64 LIDS units. We observed no statistically significant relationship between age and LIDS level (Table 4). Because there were only a few women (n = 12) in the study population, we conducted the analyses also for men only. The results were similar but a little more pronounced regarding the group effect (Table A in S2 File). Comparison of sleep quality by LIDS levels between workers with normal and extended shift length from company 1 did not show any differences (Table B in S2 File).

The most prominent effect on LIDS amplitude is exerted by shift system (Fig 1E). Workers in 12h-weekend shifts were presented with highest amplitudes implying that their ultradian rhythmicity is especially stable. Similar to LIDS levels, LIDS amplitude showed a systematic decline over the course of the sleep bout ($\beta_{\text{LIDS cycle number}}$ = -7.38, Table 4). LIDS amplitude also marginally decreased by age. With every ten years of age, amplitude decreased by 1.17 LIDS units (Table 4). In contrast to LIDS level, LIDS amplitude was neither affected by schedule-based sleep characteristics nor by sex (Fig 1F–1H).

Overall, LIDS levels and amplitude did not differ between early, intermediate or late chronotypes (Table C in S2 File). However, stratified analysis by chronotype revealed that the effect

**Table 3. Mean age-adjusted sleep duration and sleep debt including naps with 95% confidence intervals (CI) assessed with actigraphy.**

| | Total | 12h-shifts (day/night) | 8h-shifts with night work | 12h-weekend shifts (day/night) | 8h-shifts without night work | Permanent night shifts | |
|---|---|---|---|---|---|---|---|
| | N = 129 | N = 47 | N = 43 | N = 11 | N = 12 | N = 16 | |
| | Mean (95% CI) | Mean (95% CI) | Mean (95% CI) | Mean (95% CI) | Mean (95% CI) | Mean (95% CI) | F test |
| Overall duration [h][a] | 181 (177–185) | 186 (179–193) | 178 (171–186) | 183 (171–196) | 175 (162–189) | 178 (167–189) | 0.4541 |
| **Daily duration [h:min]** | | | | | | | |
| Overall | 6:26 (6:18–6:34) | 6:34 (6:21–6:47) | 6:19 (6:06–6:33) | 6:39 (6:13–7:06) | 6:16 (5:51–6:42) | 6:20 (5:57–6:42) | 0.3837 |
| **Workdays** | | | | | | | |
| Overall | 5:24 (5:15–5:32) | 5:00 (4:46–5:14) | 5:40 (5:27–5:54) | 5:13 (4:45–5:42) | 5:39 (5:13–6:04) | 5:41 (5:19–6:04) | 0.0005 |
| Morning shifts | 5:08 (4:59–5:18) | 5:02 (4:47–5:17) | 5:12 (4:56–5:28) | 5:06 (4:34–5:38) | 5:18 (4:50–5:46) | - | 0.6970 |
| Evening shifts | 6:54 (6:34–7:13) | - | 7:06 (6:45–7:27) | - | 6:13 (5:34–6:51) | - | 0.0184 |
| Night shifts | 5:10 (5:00–5:20) | 4:57 (4:41–5:12) | 5:03 (4:47–5:20) | 5:19 (4:47–5:51) | - | 5:41 (5:19–6:04) | 0.0087 |
| **Work-free days** | | | | | | | |
| Overall | 7:46 (7:33–7:59) | 8:05 (7:45–8:24) | 7:17 (6:54–7:41) | 7:53 (7:10–8:36) | 7:02 (6:17–7:46) | 8:13 (7:36–8:50) | 0.0283 |
| After morning shifts | 7:34 (7:20–7:49) | 7:44 (7:22–8:06) | 7:23 (6:58–7:47) | 7:40 (6:53–8:27) | 7:22 (6:35–8:09) | - | 0.6487 |
| After evening shifts | 6:44 (6:00–7:28) | - | 6:46 (5:51–7:41) | - | 6:41 (5:23–7:58) | - | 0.9063 |
| After night shifts | 8:10 (7:52–8:27) | 8:20 (7:56–8:44) | 7:20 (6:34–8:06) | 8:15 (7:21–9:08) | | 8:13 (7:36–8:50) | 0.1501 |
| Subsequent free days | 7:01 (6:49–7:12) | 7:23 (7:03–7:43) | 6:53 (6:34–7:12) | 7:10 (6:35–7:46) | 6:55 (6:17–7:33) | 6:19 (5:46–6:52) | 0.0257 |
| **Daily debt [h:min]** | | | | | | | |
| Overall | 2:21 (2:08–2:34) | 3:05 (2:45–3:24) | 1:41 (1:21–2:01) | 2:38 (1:58–3:18) | 1:24 (0:46–2:02) | 2:32 (1:59–3:06) | <0.0001 |
| Morning shifts | 2:26 (2:10–2:42) | 2:45 (2:20–3:10) | 2:10 (1:44–2:35) | 2:33 (1:44–3:23) | 2:06 (1:19–2:54) | - | 0.2013 |
| Evening shifts | 1:19 (0:51–1:46) | - | 1:14 (0:39–1:48) | - | 1:28 (0:39–2:17) | - | 0.6368 |
| Night shifts | 3:00 (2:42–3:17) | 3:22 (2:57–3:48) | 2:38 (2:01–3:15) | 2:56 (2:05–3:47) | | 2:32 (1:59–3:06) | 0.0818 |
| Daily nap duration at workdays [h:min] | 2:18 (2:04–2:33) | 1:44 (1:21–2:08) | 2:55 (2:34–3:16) | 1:48 (1:04–2:31) | 1:57 (1:02–2:51) | 2:15 (1:48–2:42) | 0.0005 |

[a]Overall sleep duration incl. naps only for subjects with complete actigraphy measurements over the whole study period of 28 days (n = 99).

of permanent night shifts on LIDS levels was strongest in subjects with early chronotype (Table D in S2 File). Furthermore, the decline of LIDS amplitude by age but also for different rosters was strongest in subjects with late chronotype ($\beta_{age\ per\ ten\ years}$ = -5.63, 95% CI -8.75 to -2.47, Table E in S2 File).

## Discussion

In this study, we provide new insights into the sleep characteristics of shift workers in different shift systems and shift types, including permanent night shifts and extended shifts (12h-shifts).

**Table 4. Linear mixed model analysis for LIDS levels and range of LIDS oscillation.**

| | LIDS level | | | LIDS oscillation | | |
|---|---|---|---|---|---|---|
| Random effects | | Variance | SD | | Variance | SD |
| Subject (Sleep bout) | Intercept | 76.12 | 8.73 | | 9.68 | 3.11 |
| Subject | Intercept | 110.54 | 10.51 | | 28.33 | 5.32 |
| | LIDS cycle | 9.95 | 3.15 | | 2.79 | 1.67 |
| Residual | | 630.23 | 25.10 | | 462.190 | 21.499 |
| Fixed effects | β | 95% CI | | β | 95% CI | |
| Intercept | 73.44 | 61.55 | 85.31 | 77.62 | 67.60 | 87.63 |
| Age (10 years) | -1.20 | -3.05 | 0.64 | -1.17 | -2.47 | 0.13 |
| Men[a] | -7.65 | -14.35 | -0.95 | -3.11 | -8.02 | 1.78 |
| Sleep duration (h) | -0.86 | -1.14 | -0.58 | 0.34 | -0.27 | 0.95 |
| LIDS cycle number | -12.63 | -16.25 | -9.00 | -7.38 | -11.24 | -3.51 |
| Night sleep[b] | -6.78 | -9.36 | -4.21 | 0.16 | -3.57 | 3.85 |
| Work day[c] | -1.29 | -2.36 | -0.21 | 2.42 | -0.05 | 4.87 |
| Roster[d] | | | | | | |
| 12h-shifts (day/night) | -5.66 | -12.83 | 1.51 | -5.54 | -10.57 | -0.50 |
| 8h-shifts with night work | -6.19 | -13.35 | 0.99 | -7.55 | -12.60 | -2.50 |
| 8h-shifts without night work | -1.32 | -10.06 | 7.43 | -5.62 | -11.84 | 0.62 |
| Permanent night shifts | -10.07 | -18.23 | -1.91 | -4.76 | -10.66 | 1.12 |
| Interaction effects | | | | | | |
| Men[a]*night sleep[b] | 3.66 | 1.04 | 6.29 | 2.13 | -1.21 | 5.44 |
| LIDS cycle number*age | 0.40 | -0.16 | 0.96 | -0.01 | -0.47 | 0.46 |
| LIDS cycle number*men[a] | 0.25 | -1.74 | 2.24 | 1.08 | -0.60 | 2.78 |
| LIDS cycle number*sleep duration | 0.64 | 0.53 | 0.75 | 0.34 | 0.08 | 0.61 |
| LIDS cycle number*night sleep[b] | 1.43 | 1.07 | 1.79 | -0.44 | -1.32 | 0.47 |
| LIDS cycle number*work day[c] | 0.94 | 0.54 | 1.35 | -0.79 | -1.85 | 0.27 |
| LIDS cycle number*roster[d] | | | | | | |
| 12h-shifts (day/night) | 1.65 | -0.52 | 3.81 | 0.49 | -1.32 | 2.29 |
| 8h-shifts with night work | 1.59 | -0.57 | 3.76 | 1.03 | -0.79 | 2.85 |
| 8h-shifts without night work | -0.06 | -2.71 | 2.58 | 0.88 | -1.39 | 3.15 |
| Permanent night shifts | 3.09 | 0.61 | 5.56 | 0.52 | -1.66 | 2.70 |

SD = standard deviation; CI = 95% confidence interval

[a]reference: women

[b]reference: no sleep between 8pm and 8am

[c]reference: unrestricted sleep episodes

[d]reference: 12h-weekend shifts (day/night).

Over the long study period of 28 days, we did not observe any sleep differences among shift workers of different rosters. However, partial sleep deprivation was particularly evident in rotational shift systems with 12h-shifts. Here, sleep duration on workdays was particularly low, while overall sleep debt was highest. Additionally, absolute social jetlag was also higher in 12h-shift workers compared with subjects working normal 8h-shifts with night work. Overall, permanent night workers had the highest social jetlag and latest mid-sleeps on workdays and free days. Sleep quality assessed with LIDS was particularly reduced in permanent night shift workers and differed between daytime and nighttime sleep.

We found a median daily sleep duration of 6:26 h:min which is in the same range as observed in workers in a fast-rotating 12h-shift schedule with an overall sleep duration of 6:49

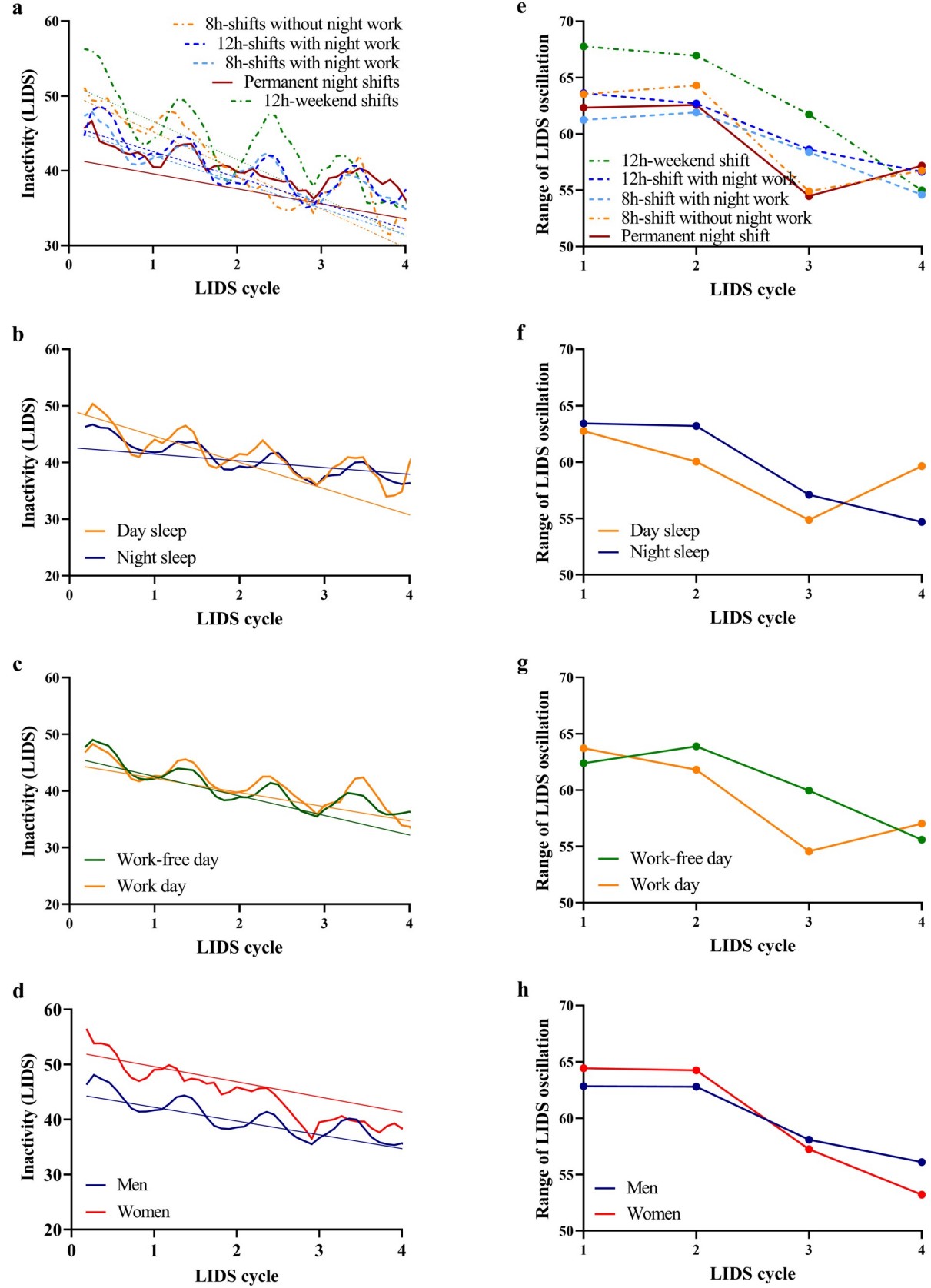

**Fig 1. Locomotor inactivity during sleep (LIDS) levels and range of LIDS oscillation per LIDS cycle across groups.** Average raw LIDS profiles and linear fits assessed with mixed models as described in the method section on the left and range of LIDS oscillation on the right for indicated groups: roster (a/e), daytime and nighttime sleep (b/f), work-free and work day (c/g), sex (d/h).

h:min [27]. We also determined a similar acute sleep loss especially in workers with 12h-shifts including night work who were presented with the shortest main sleep after night shifts (<5h) and suffered from a daily sleep debt of about three hours. In general, sleep debts were highest during night shifts and lowest during evening shifts confirming results from a meta-analysis based on self-reported sleep durations [3]. Nevertheless, extremely short (<6h) but also extremely long (>9h) sleep durations are linked to increased health risks [29] which we have observed for 26% respectively 1% of night sleep bouts within this study.

In contrast to a recently published study reporting that sleeping during daytime on both workdays and free days was a commonly used sleep strategy during night shifts for about 20% of hospital shift workers [30], we did not observe a general adaption of permanent night shift workers to the imposed work schedule which is consistent with studies examining circadian adaptation to permanent night shifts measured by objective phase markers [31]. Here, social jetlag was largest in permanent night shift workers, who cannot compensate severe social jetlag on night shifts with weaker effects from other shift types. Accordingly, permanent night shift workers have much more pronounced problems adjusting to the overall social time on work-free days than shift workers of other rosters. Hence, they may not benefit over rotating shift systems regarding fewer health and safety problems.

In this study, midpoint of sleep and chronotype were estimated by actimetry and revealed a mid-sleep between two work-free days at 4:12 being very similar to the general population with 4:14 [32]. Evidence from the current study and from the literature indicate that chronotype differ between rosters [6, 27, 33]. For instance, we observed the earliest chronotype for workers without night work (mean 3:30, 95% CI 2:26–3:36) which is similar to the reported chronotype of 3:18 for shift workers in a poultry-processing company in Brazil [33]. Closely related to the chronotype concept is social jetlag depicting the difference between social and biological timing [28]. In line with earlier studies [6, 33], we observed different social jetlag between rosters and work shifts with highest social jetlag of 5h in permanent night workers.

We are the first to link chronotype with sleep structure assessed with the LIDS approach. Earlier attempts to elucidate this relationship concentrated on questionnaire data about sleep disturbance or simplistic sleep quality measures such as the percentage of time in bed spent actually sleeping [6, 34]. In this study, we observed that extreme chronotypes (early and late) showed a stronger impact of roster on sleep quality than intermediate chronotypes. In detail, early chronotypes showed a stronger decrease of LIDS for workers with 8h-shifts including night work or permanent night shift workers. On the other hand, late chronotypes exhibited less stable LIDS cycles, especially when working permanent night shifts. This is in line with earlier findings of late chronotypes being more affected by poor sleep quality [35]. In general, working in permanent night shifts revealed the lowest LIDS levels. In contrast, working 12h-weekend shifts, i.e. 24 weekly working hours, were presented with highest LIDS amplitudes implying an especially stable ultradian rhythmicity. Furthermore, our findings support previous results that women move less during sleep than men [24]. This study verifies that this applies also to shift workers. On the contrary to earlier results where age was the most prominent effect [24], we did not observe age specific differences in LIDS levels or LIDS decline which might be due to the small age range of the participants (median 47 years, IQR 41–54, range 20–64).

Overall, napping was common within this study which confirms earlier observations from a study of shift workers [27]. Here, during workdays, workers with permanent night shifts and workers with 8h-shifts including night work compensated for shorter main sleeps with extensive naps. Hence, sleep debt was reduced which has been reported by an meta-analysis in night-shift workers before [36].

Some limitations of our study are noteworthy. In spite of the large sample size of this cross-sectional study examining shift workers some rosters were presented only by a few subjects, limiting the statistical power and the drawing of conclusions. Along with residual confounding inherent to field studies the generalizability of our findings might be impaired. It might be possible that shift workers were assigned to a specific shift system so that selection bias cannot be ruled out. We assessed sleep structure with the innovative approach of limb movement during sleep measured via actigraphy, which is easily applicable in a large number of subjects with many sleep episodes per subject. However, we did not verify our results with established methods such as PSG.

## Conclusions

This study confirms that partial sleep deprivation is a common consequence of shift work. Very high daily sleep debts were observed for long working hours in fast-rotating 12h-shifts, especially after night shifts. Additionally, working 12h-shifts or permanent night shifts resulted in strong absolute social jetlag. Sleep quality was especially reduced in permanent night shift workers compared with shift workers of other rosters.

## Supporting information

**S1 File. Representation of shift schedules of the different rosters.** M = morning shift; E = evening shift; N = night shift; D = day shift.
(DOCX)

**S2 File. Additional linear mixed model analysis for LIDS levels and range of LIDS oscillation.**
(DOCX)

## Acknowledgments

We thank Simone Putzke (IPA) for excellent technical assistance and prudent data management as well as the employees of ars serendi for their field investigations.

## Author Contributions

**Conceptualization:** Frank Brenscheidt, Sylvia Rabstein.

**Data curation:** Frank Brenscheidt, Anita Tisch.

**Formal analysis:** Swaantje Casjens.

**Writing – original draft:** Swaantje Casjens.

**Writing – review & editing:** Frank Brenscheidt, Anita Tisch, Beate Beermann, Thomas Brüning, Thomas Behrens, Sylvia Rabstein.

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
