## [Decision Letter · Decision Letter 0]

7 Oct 2021

PONE-D-21-23813Sleep of shift workers in different shift-work systems assessed with actigraphyPLOS ONE

Dear Dr. Casjens,

Thank you for submitting your manuscript to PLOS ONE. After careful consideration, we feel that it has merit but does not fully meet PLOS ONE’s publication criteria as it currently stands. Therefore, we invite you to submit a revised version of the manuscript that addresses the points raised during the review process.

The reviewers have requested some minor revisions and points of clarification. Please respond to all of the reviewers' comments and resubmit a revised manuscript.

We look forward to receiving your revised manuscript.

Kind regards,

Ben Bullock

Academic Editor

PLOS ONE

Journal Requirements:

2. PLOS ONE does not permit references to unpublished data; therefore, we request that you either include the referenced data or remove the instances of "data not shown," "unpublished results," or similar

3. Please include additional information regarding the survey or questionnaire used in the study and ensure that you have provided sufficient details that others could replicate the analyses. For instance, if you developed a questionnaire as part of this study and it is not under a copyright more restrictive than CC-BY, please include a copy, in both the original language and English, as Supporting Information

"We also acknowledge the support by the Open Access Publication Funds of the Ruhr-Universität Bochum."

"The authors received no specific funding for this work"

6. We note that you have included the phrase “data not shown” in your manuscript. Unfortunately, this does not meet our data sharing requirements. PLOS does not permit references to inaccessible data. We require that authors provide all relevant data within the paper, Supporting Information files, or in an acceptable, public repository. Please add a citation to support this phrase or upload the data that corresponds with these findings to a stable repository (such as Figshare or Dryad) and provide and URLs, DOIs, or accession numbers that may be used to access these data. Or, if the data are not a core part of the research being presented in your study, we ask that you remove the phrase that refers to these data

Reviewers' comments:

Reviewer's Responses to Questions

**Comments to the Author**

1. Is the manuscript technically sound, and do the data support the conclusions?

Reviewer #1: Yes

Reviewer #2: Partly

2. Has the statistical analysis been performed appropriately and rigorously? 

Reviewer #1: Yes

Reviewer #2: Yes

3. Have the authors made all data underlying the findings in their manuscript fully available?

Reviewer #1: Yes

Reviewer #2: No

4. Is the manuscript presented in an intelligible fashion and written in standard English?

Reviewer #1: Yes

Reviewer #2: Yes

5. Review Comments to the Author

Reviewer #1: This paper has focus on the association of different shift systems with sleep length and the structure of sleep both during work and free days, with a long follow-up for an experimental study. It investigated the association of five shift rosters on sleep using wrist actigraphs over 28 days. The paper adds to the current knowledge due to having indeed the scope on the whole shift roster. Up to now most earlier papers focus on shift-dependent associations, but often forget the significance of free days in the rosters for recovery, sleep and the also to the related overall circadian disruption influencing the health effects of shift work. The paper is also valuable due to having several well-established objective outcomes: sleep timing, duration, overall sleep debt, the quality of sleep (LIDS) and social jetlag – the latest giving indications on the circadian disruption that is probably most relevant for the long-term effects of shift work on health. Indeed, the paper has several original and interesting results, as described in the abstract and the first chapter of the discussion.

The sample is cross-sectional but large enough to study the five rosters – that are used generally also in other countries than Germany, increasing the value of the paper. Especially the results of the permanent night work and those of the quickly and slowly rotating shift rotas are interesting. The rosters are well described in the appendix.

The paper included many results, but they are comprehensive and instead of publishing each outcome in a different journal, bringing them all to one paper (if the paper limits allow this) supports the overall interpretation of the message.

Minor issues

- the title is a bit boring compared to the contents – could reflect more the added value of the paper

- abstract: methods: normally past tense is used “we investigated…” etc. Some sentences could also benefit from language revision to clarify the meanings

- did you select the most relevant results to the conclusions here (why focus on 12-hour shifts only)– compare to the conclusions of the main text that gives, I think, a better view of the key results

- introduction includes relevant literature

- material and methods: lines 171-172: do the definitions of day sleep and night sleep include all options? For example, how a sleep period starting at 03 am and ending at 11 am is scored?

- results: line 248 “experienced” hints to subjective data – did it come from a survey” I would say that workers with .. had the shortest shift work experience

- lines 265-267 explanation for the results (“cannot compensate”) should be in the discussion, not here – an interesting topic to discuss

- line 343: how did it differ? Please report in a way that we could read in which shift schedule the sleep quality was good, where it was bad. “Differed” is not too informative yet.

Reviewer #2: Main comments to authors:

The authors conclude in the abstract that “this study confirms that 12h-shifts are not associated with chronic sleep loss compared to 8h-shifts, when observing long time frames of four weeks”. Then in the main text conclusion “This study confirms that partial sleep deprivation but not chronic sleep loss is a common consequence of shift work”. I do not see where this rather strong conclusion is drawn from. The sleep durations on work days are <7 hours on average in all shift rotation conditions. Both schedules have chronic sleep loss - from the abstract conclusion is the point that both shift schedules are just as bad as each other? There is evidence from numerous laboratory studies that sleep restriction builds over time, and is not sufficiently “repaid” by 1-2 long recovery sleeps, with performance deficits showing a cumulative effect of chronic sleep loss. How, based on the data presented in this manuscript, is there evidence of no chronic sleep loss associated with shift work? This seems to go beyond what the data presented can determine.

“Circadian misalignment was quantified by social jetlag”. Social jetlag is not necessarily a measure of circadian misalignment, as you have no objective marker rhythms measured (e.g., melatonin). While social jetlag is sometimes used as a proxy, it is important to acknowledge that while it may indicate possible misalignment, it is not a direct measure of the clock, rather a sleep-based marker. Sleep can be misaligned with endogenous circadian rhythms, with wide inter-individual variation, particularly in shift workers. There are multiple reasons why shift workers sleep at different times, that are not all attributable to circadian disruption. I do not see how social jetlag can be an accurate measure of circadian disruption in this context. Rather it is a marker of how disrupted sleep timing is compared to free days.

During the sleep analysis, were the sleep on- and offsets derived from the device software compared with sleep diaries for any major discrepancies? This is a commonly used process to eliminate any obvious errors that can occur due to non-wear or individuals being still during wake times. Similarly, how were naps measured? Were naps derived using actigraphy only, or alongside sleep diaries? Shorter bouts of inactivity (e.g., watching a movie) can easily be misinterpreted as a nap when relying solely on actigraphy. Given the high rates of naps in this dataset, and the presence of both wrist actigraphy and sleep diaries, I would like more detail on how these were cross-checked.

Discussion line 365: The observation that permanent night shift workers did not adapt to the schedule in between work days is consistent with other studies examining circadian adaptation to permanent night shift schedules measured via objective phase markers, and also to behavioural/sleep observations similar to this study. I suggest including comparison or reference to this broader literature.

Discussion line 362: Chronotype is a term that is frequently misattributed, muddying the scientific literature of circadian timing, diurnal preference, and sleep timing. Chronotype does not refer to biological measurements of endogenous circadian timing such as melatonin or body temperature - it is a false equivalency to say that these measures are the same as sleep timing measures such as actigraphy or self-report questionnaires. The studies mentioned in this paragraph of the discussion are chronotype studies reporting sleep-derived metrics, not circadian phase measured via melatonin. The authors should either include studies that have examined changes in circadian timing in shift workers (e.g., Deacon & Arendt, 1996; Gibbs et al., 2007; Hansen et al., 2010; Stone et al 2018) and clarify terminology making the distinction clear, or remove reference to biological measurements in this paragraph.

Minor suggestions:

A minor request: could the authors consider reporting sleep durations in hours and minutes rather than minutes? In some places it is in hh:mm and others in minutes. I believe hh:mm is easier to interpret, though it is your choice.

Table 3 Overall duration [h] - should this be [min]?

“Earlier attempts to elucidate this relationship concentrated on … naïve sleep quality measures such as the percentage of time in bed spent actually sleeping” Why is this a naïve sleep quality measure? Naïve suggests there's a lack of experience, wisdom, or judgement in using sleep efficiency calculations as sleep quality (which is widely done in the sleep field). I'm curious about this statement? It could be interpreted as inflammatory, which I do not believe is the author's intention. Perhaps a different word might be better? Maybe the word "simplistic" might be a less contentious word choice?

Does the LIDS measure give any insight into when participants are sleeping between night shifts? For example, are individuals sleeping immediately after a shift, or remaining awake and sleeping before the next shift, are the different phenotypes? This is not critical to examine or include, but if these data are readily available it would be a valuable addition, as this is often assumed but not necessarily well described in the literature.

6. PLOS authors have the option to publish the peer review history of their article (what does this mean?). If published, this will include your full peer review and any attached files.

Reviewer #1: No

Reviewer #2: No

---

## [Author Response · Author response to Decision Letter 0]

16 Nov 2021

We thank the editor and reviewers for their comments and suggestions. Below you will find our detailed responses, which we have also uploaded in a separate file.

Journal Requirements:

Response: We have checked our manuscript and have made adjustments where necessary.

2. PLOS ONE does not permit references to unpublished data; therefore, we request that you either include the referenced data or remove the instances of "data not shown," "unpublished results," or similar

Response: We have removed all references to unpublished data. In two cases we have added supporting information (Tables B and C in S2 File).

3. Please include additional information regarding the survey or questionnaire used in the study and ensure that you have provided sufficient details that others could replicate the analyses. For instance, if you developed a questionnaire as part of this study and it is not under a copyright more restrictive than CC-BY, please include a copy, in both the original language and English, as Supporting Information

Response: We did not develop or use a specific questionnaire for this study. In particular, the extensive actigraphy data were used. The other data used for this analysis were age, gender, body mass index, years of shift work, and informatioin on the shift schedule and working hours as described in the method section. As indicated in the data availability statement, the data are available upon reasonable request from the Federal Institute for Occupational Safety and Health. The same applies to further information on the study.

Response: The support of the Open Access Publication Fund of the Ruhr-Universität Bochum mentioned in the Acknowledgements Section is a partial transfer of the publication costs for this work. The funders had no role in study design, data collection and analysis, decision to publish, or preparation of the manuscript. We have included our amended Financial Disclosure statement in the cover letter. It now reads:

“The Open Access Publication Fund of the Ruhr-Universität Bochum contributed to publication costs for this manuscript. The funders had no role in study design, data collection and analysis, decision to publish, or preparation of the manuscript.“

"We also acknowledge the support by the Open Access Publication Funds of the Ruhr-Universität Bochum."

"The authors received no specific funding for this work"

Response: We have ommitted the funding information in the Acknowledgments Section and have included our amended financial disclosure statement in the cover letter.

6. We note that you have included the phrase “data not shown” in your manuscript. Unfortunately, this does not meet our data sharing requirements. PLOS does not permit references to inaccessible data. We require that authors provide all relevant data within the paper, Supporting Information files, or in an acceptable, public repository. Please add a citation to support this phrase or upload the data that corresponds with these findings to a stable repository (such as Figshare or Dryad) and provide and URLs, DOIs, or accession numbers that may be used to access these data. Or, if the data are not a core part of the research being presented in your study, we ask that you remove the phrase that refers to these data

Response: We have removed all references to unpublished data. In two cases, we have added supporting information (S2 Table and S3 Table).

Response: We have checked and adjusted our reference list accordingly.

Reviewers' comments:

Reviewer #1: This paper has focus on the association of different shift systems with sleep length and the structure of sleep both during work and free days, with a long follow-up for an experimental study. It investigated the association of five shift rosters on sleep using wrist actigraphs over 28 days. The paper adds to the current knowledge due to having indeed the scope on the whole shift roster. Up to now most earlier papers focus on shift-dependent associations, but often forget the significance of free days in the rosters for recovery, sleep and the also to the related overall circadian disruption influencing the health effects of shift work. The paper is also valuable due to having several well-established objective outcomes: sleep timing, duration, overall sleep debt, the quality of sleep (LIDS) and social jetlag – the latest giving indications on the circadian disruption that is probably most relevant for the long-term effects of shift work on health. Indeed, the paper has several original and interesting results, as described in the abstract and the first chapter of the discussion.

The sample is cross-sectional but large enough to study the five rosters – that are used generally also in other countries than Germany, increasing the value of the paper. Especially the results of the permanent night work and those of the quickly and slowly rotating shift rotas are interesting. The rosters are well described in the appendix.

The paper included many results, but they are comprehensive and instead of publishing each outcome in a different journal, bringing them all to one paper (if the paper limits allow this) supports the overall interpretation of the message.

Minor issues

1. The title is a bit boring compared to the contents – could reflect more the added value of the paper

Response: In fact, the original title was a bit boring. We have changed it to:

Social jetlag and sleep debts are altered in different rosters of night shift work

2. Abstract: 

• Methods: normally past tense is used “we investigated…” etc. Some sentences could also benefit from language revision to clarify the meanings.

Response: We have changed the tense accordingly.

• Did you select the most relevant results to the conclusions here (why focus on 12-hour shifts only) - compare to the conclusions of the main text that gives, I think, a better view of the key results.

Response: We have adjusted the conclusions, taking also the comments of the other reviewer into account. It now reads: 

„Shift work leads to partial sleep deprivation, which particularly affects workers in 12h-shifts or permanent night shifts. Working these shifts resulted in higher sleep debts and larger absolute social jetlag whereas sleep quality was especially reduced in permanent night shift workers compared with shift workers of other rosters.“

3. Introduction includes relevant literature

Response: Thank you

4. Material and methods: Lines 171-172: do the definitions of day sleep and night sleep include all options? For example, how a sleep period starting at 03 am and ending at 11 am is scored?

Response: In the general analysis of the sleep data, we considered all sleep episodes. When dividing into day and night sleep, we used the mentioned boundaries to have these strictly defined. Hence, a sleep episode starting at 03 am and ending at 11 am felt into a third group of unclassified episodes which we have now also mentioned in the method section.

5. Results: 

• Line 248 “experienced” hints to subjective data – did it come from a survey” I would say that workers with .. had the shortest shift work experience

Response: We have adjusted the wording accordingly.

- Lines 265-267 explanation for the results (“cannot compensate”) should be in the discussion, not here – an interesting topic to discuss

Response: We have removed the relevant sentence from the results section and inserted it in the associated paragraph in the discussion (lines 375-379 in the version with track changes).

- Line 343: how did it differ? Please report in a way that we could read in which shift schedule the sleep quality was good, where it was bad. “Differed” is not too informative yet.

Response: We have adjusted the corresponding sentence as follows: „Sleep quality assessed with LIDS was paricularly reduced in permanent night shift workers and differed between daytime and nighttime sleep.”

Reviewer #2: Main comments to authors:

The authors conclude in the abstract that “this study confirms that 12h-shifts are not associated with chronic sleep loss compared to 8h-shifts, when observing long time frames of four weeks”. Then in the main text conclusion “This study confirms that partial sleep deprivation but not chronic sleep loss is a common consequence of shift work”. I do not see where this rather strong conclusion is drawn from. The sleep durations on work days are <7 hours on average in all shift rotation conditions. Both schedules have chronic sleep loss - from the abstract conclusion is the point that both shift schedules are just as bad as each other? There is evidence from numerous laboratory studies that sleep restriction builds over time, and is not sufficiently “repaid” by 1-2 long recovery sleeps, with performance deficits showing a cumulative effect of chronic sleep loss. How, based on the data presented in this manuscript, is there evidence of no chronic sleep loss associated with shift work? This seems to go beyond what the data presented can determine.

Response: With our statement about chronic sleep loss, we were referring to the nonexistent difference between the rosters in terms of the overall sleep duration over the whole study period of 28 days as well as the overall daily sleep duration as shown in Table 3. We agree that sleep impairment cannot be accommodated by a few long sleep episodes. Hence, we have now ommited the term "chronic sleep loss" throughout the manuscript.

“Circadian misalignment was quantified by social jetlag”. Social jetlag is not necessarily a measure of circadian misalignment, as you have no objective marker rhythms measured (e.g., melatonin). While social jetlag is sometimes used as a proxy, it is important to acknowledge that while it may indicate possible misalignment, it is not a direct measure of the clock, rather a sleep-based marker. Sleep can be misaligned with endogenous circadian rhythms, with wide inter-individual variation, particularly in shift workers. There are multiple reasons why shift workers sleep at different times, that are not all attributable to circadian disruption. I do not see how social jetlag can be an accurate measure of circadian disruption in this context. Rather it is a marker of how disrupted sleep timing is compared to free days.

Response: Indeed, the statement that circadian misalignment can be measured by means of social jetlag is too concise. We have taken up your suggestion and have now written in the text that social jetlag is used as a proxy for circadian misalignment and that increased social jetlag values indicate a greater disrupted sleep timing between workdays and work-free days (lines 195-200 in the version with track changes).

During the sleep analysis, were the sleep on- and offsets derived from the device software compared with sleep diaries for any major discrepancies? This is a commonly used process to eliminate any obvious errors that can occur due to non-wear or individuals being still during wake times. Similarly, how were naps measured? Were naps derived using actigraphy only, or alongside sleep diaries? Shorter bouts of inactivity (e.g., watching a movie) can easily be misinterpreted as a nap when relying solely on actigraphy. Given the high rates of naps in this dataset, and the presence of both wrist actigraphy and sleep diaries, I would like more detail on how these were cross-checked.

Response: Two authors (SC and SR) have cross-checked all sleep on- and offsets derived from actigraphy and from sleep diaries for each participant. Indeed, there were differences especially concerning naps. By means of expert rating, we decided in individual cases whether sleep was actually present or not. Furthermore, the diaries also documented the discarding of the actigraphs, so that we could also clarify falsely "measured" sleep episodes through the comparison. 

We have edited the corresponding paragraph in the methods section accordingly (lines 158-162 in the version with track changes).

Discussion line 365: The observation that permanent night shift workers did not adapt to the schedule in between work days is consistent with other studies examining circadian adaptation to permanent night shift schedules measured via objective phase markers, and also to behavioural/sleep observations similar to this study. I suggest including comparison or reference to this broader literature.

Response: We have added two references to the discussion at this point:

Garde AH et al. How to schedule night shift work in order to reduce health and safety risks. Scand J Work Environ Health. 2020; 46:557–69. https://doi.org/10.5271/sjweh.3920

Harrison EM at al. Sleep-Scheduling Strategies in Hospital Shiftworkers. Nat Sci Sleep. 2021; 13:1593–609. https://doi.org/10.2147/NSS.S321960

Discussion line 362: Chronotype is a term that is frequently misattributed, muddying the scientific literature of circadian timing, diurnal preference, and sleep timing. Chronotype does not refer to biological measurements of endogenous circadian timing such as melatonin or body temperature - it is a false equivalency to say that these measures are the same as sleep timing measures such as actigraphy or self-report questionnaires. The studies mentioned in this paragraph of the discussion are chronotype studies reporting sleep-derived metrics, not circadian phase measured via melatonin. The authors should either include studies that have examined changes in circadian timing in shift workers (e.g., Deacon & Arendt, 1996; Gibbs et al., 2007; Hansen et al., 2010; Stone et al 2018) and clarify terminology making the distinction clear, or remove reference to biological measurements in this paragraph.

Response: Chronotype is considered only as a secondary factor in this publication. For example, all tables describing a possible association between LIDS and chronotype are in the Supplement. Therefore, we prefer not to clarify terminology at this point, but limit ourselves to studies that record sleep-related measures.

We have edited the paragraph accordingly.

Minor suggestions:

A minor request: could the authors consider reporting sleep durations in hours and minutes rather than minutes? In some places it is in hh:mm and others in minutes. I believe hh:mm is easier to interpret, though it is your choice.

Table 3 Overall duration [h] - should this be [min]?

Response: We have adjusted all corresponding data in the manuscript as well as the duration of the social jetlag in Table 2 and the data in Table 3. The overall mean age-adjusted sleep duration in Table 3 depicts the overall sleep duration including naps over the whole study period of 28 days. Therefore, the unit h is correct. This shows that the total sleep over a month does not differ between the shift systems.

“Earlier attempts to elucidate this relationship concentrated on … naïve sleep quality measures such as the percentage of time in bed spent actually sleeping” Why is this a naïve sleep quality measure? Naïve suggests there's a lack of experience, wisdom, or judgement in using sleep efficiency calculations as sleep quality (which is widely done in the sleep field). I'm curious about this statement? It could be interpreted as inflammatory, which I do not believe is the author's intention. Perhaps a different word might be better? Maybe the word "simplistic" might be a less contentious word choice?

Response: Indeed, the choice of words was unfavorable and a devaluation was not intended in any respect. We gladly accept your suggestion and have replaced "naive" by "simplistic".

Does the LIDS measure give any insight into when participants are sleeping between night shifts? For example, are individuals sleeping immediately after a shift, or remaining awake and sleeping before the next shift, are the different phenotypes? This is not critical to examine or include, but if these data are readily available it would be a valuable addition, as this is often assumed but not necessarily well described in the literature.

Response: The LIDS measure represents a proxy for sleep quality independent of the time of sleep. In our analysis, for example, we were able to show that sleep quality varied between day and night sleep. However, the present analysis did not include whether the participant slept immediately following the shift or just before the next shift. This is indeed an interesting question that should be explored in a future study or possibly in a reanalysis of the present data.

---

## [Decision Letter · Decision Letter 1]

16 Dec 2021

Social jetlag and sleep debts are altered in different rosters of night shift work

PONE-D-21-23813R1

Dear Dr. Casjens,

We’re pleased to inform you that your manuscript has been judged scientifically suitable for publication and will be formally accepted for publication once it meets all outstanding technical requirements.

Kind regards,

Ben Bullock

Academic Editor

PLOS ONE

Additional Editor Comments (optional):

Reviewers' comments:

Reviewer's Responses to Questions

**Comments to the Author**

1. If the authors have adequately addressed your comments raised in a previous round of review and you feel that this manuscript is now acceptable for publication, you may indicate that here to bypass the “Comments to the Author” section, enter your conflict of interest statement in the “Confidential to Editor” section, and submit your "Accept" recommendation.

Reviewer #1: All comments have been addressed

Reviewer #2: All comments have been addressed

2. Is the manuscript technically sound, and do the data support the conclusions?

Reviewer #1: Yes

Reviewer #2: (No Response)

3. Has the statistical analysis been performed appropriately and rigorously? 

Reviewer #1: Yes

Reviewer #2: (No Response)

4. Have the authors made all data underlying the findings in their manuscript fully available?

Reviewer #1: Yes

Reviewer #2: (No Response)

5. Is the manuscript presented in an intelligible fashion and written in standard English?

Reviewer #1: (No Response)

Reviewer #2: (No Response)

6. Review Comments to the Author

Reviewer #1: The authors have addressed the referee comments well.

Reviewer #2: (No Response)

7. PLOS authors have the option to publish the peer review history of their article (what does this mean?). If published, this will include your full peer review and any attached files.

Reviewer #1: **Yes: **Mikko Härmä

Reviewer #2: No

---

## [Editor Report · Acceptance letter]

31 Dec 2021

PONE-D-21-23813R1 

Social jetlag and sleep debts are altered in different rosters of night shift work 

Dear Dr. Casjens:

I'm pleased to inform you that your manuscript has been deemed suitable for publication in PLOS ONE. Congratulations! Your manuscript is now with our production department. 

Kind regards, 

on behalf of

Dr. Ben Bullock 

Academic Editor

PLOS ONE